# RgsA Attenuates the PKA Signaling, Stress Response, and Virulence in the Human Opportunistic Pathogen *Aspergillus fumigatus*

**DOI:** 10.3390/ijms20225628

**Published:** 2019-11-11

**Authors:** Hnin Phyu Lwin, Yong-Ho Choi, Min-Woo Lee, Jae-Hyuk Yu, Kwang-Soo Shin

**Affiliations:** 1Department of Microbiology, Graduate School, Daejeon University, Daejeon 34520, Korea; hninphyuyau@gmail.com (H.P.L.); youngho1107@gmail.com (Y.-H.C.); 2Soonchunhyang Institute of Medi-bio Science, Soonchunhyang University, Chungcheongnam-do 31151, Korea; mwlee12@sch.ac.kr; 3Department of Bacteriology, University of Wisconsin-Madison, Madison, WI 53706, USA; 4Department of Systems Biotechnology, Konkuk University, Seoul 05029, Korea

**Keywords:** human pathogenic fungi, *Aspergillus fumigatus*, RGS, development, stress response, virulence, transcriptomic analysis

## Abstract

The regulator of G-protein signaling (RGS) proteins play an important role in upstream control of heterotrimeric G-protein signaling pathways. In the genome of the human opportunistic pathogenic fungus *Aspergillus fumigatus*, six RGS protein-encoding genes are present. To characterize the *rgsA* gene predicted to encode a protein with an RGS domain, we generated an *rgsA* null mutant and observed the phenotypes of the mutant. The deletion (Δ) of *rgsA* resulted in increased radial growth and enhanced asexual sporulation in both solid and liquid culture conditions. Accordingly, transcripts levels of the key asexual developmental regulators *abaA*, *brlA,* and *wetA* are elevated in the Δ*rgsA* mutant. Moreover, Δ*rgsA* resulted in elevated spore germination rates in the absence of a carbon source. The activity of cAMP-dependent protein kinase A (PKA) and mRNA levels of genes encoding PKA signaling elements are elevated by Δ*rgsA*. In addition, mRNA levels of genes associated with stress-response signaling increased with the lack of *rgsA*, and the Δ*rgsA* spores showed enhanced tolerance against oxidative stressors. Comparative transcriptomic analyses revealed that the Δ*rgsA* mutant showed higher mRNA levels of gliotoxin (GT) biosynthetic genes. Accordingly, the *rgsA* null mutant exhibited increased production of GT and elevated virulence in the mouse. Conversely, the majority of genes encoding glucan degrading enzymes were down-regulated by Δ*rgsA,* and endoglucanase activities were reduced. In summary, RgsA plays multiple roles, governing growth, development, stress responses, virulence, and external polymer degradation—likely by attenuating PKA signaling.

## 1. Introduction

*Aspergillus fumigatus* is a widespread saprophytic fungus in nature, typically found in soil and decaying vegetation [1,2]. This ubiquitous fungus is the most prevalent airborne fungal pathogen, causing a multitude of diseases in humans, such as allergic bronchopulmonary aspergillosis, aspergilloma, and invasive aspergillosis [3]. Most importantly, as an opportunistic human pathogen, *A*. *fumigatus* can cause a serious invasive pulmonary aspergillosis leading to an approximately 90% mortality rate, mainly in immunocompromised patients [4,5]. However, in healthy individuals, mucocilliary clearance and pulmonary immune defenses clear several hundred spores inhaled daily [6].

Heterotrimeric G-protein (G-protein)-mediated signaling pathways play an important role in the regulation of asexual sporulation, vegetative growth, and sensing various extracellular signals in filamentous fungi [7,8]. A canonical G-protein signaling system consists of a G-protein coupled receptor (GPCR); Gα, β, and γ subunits; and a variety of effector proteins [9,10,11]. In various fungi, regulators of G-protein signaling (RGSs) have been shown to regulate asexual differentiation, secondary metabolism, and pathogenicity [12,13,14,15,16,17]. Characterizations of the roles of RGS proteins may serve as the bases for the identification of novel targets to control human pathogenic fungi.

In the *A*. *fumigatus* genome, six genes are predicted to encode RGSs: *flbA*, *gprK*, *rgsA*, *rax1* (*rgsB*), *rgsC*, and *rgsD*. Previously we characterized all but RgsA, of those RGS proteins [13,15,16,17,18]. RgsA is similar to *Saccharomyces cerevisiae* Rgs2p [19], which has the RGS domain at the N-terminus and represents the second and specific RGS protein. The deletion of *rgs2* reduced thermal tolerance and overexpression of *rgs2* caused a significant increase in temperature resistance. In contrast, the deletion of *rgsA* in *Aspergillus nidulans* resulted in elevated mycelial and conidial pigmentation, and increased the resistance of conidia and vegetative hyphae to oxidative and thermal stresses [20]. As the ability to cope with various external stresses is one of the key survival factors of *A. fumigatus*, we hypothesized that the *A. fumigatus rgsA* gene is involved in proper cellular responses to environmental changes. This study presents a thorough functional characterization of RgsA’s governing of various key biological processes in *A. fumigatus*.

## 2. Results

### 2.1. Summary of A. fumigatus RgsA

RgsA (Afu6g06860), a putative regulator of G-protein signaling, has a typical RGS domain. The *rgsA* gene maps to chromosome VI and its ORF consists of 1104 bp of nucleotides with no introns, leading to a 368-amino-acid length protein. The domain structure of RgsA is very simple and contains three low complexity domains (21 to 48 aa, 248 to 263 aa, and 306 to 327 aa), and an RGS domain (66 to 206 aa, E-value: 4.86e-17; Appendix A). With these protein sequences, we compared with RgsA-like proteins in other aspergilli (Appendix A). The *A*. *fumigatus* RgsA protein is closely related to that of *Neosartorya fischeri*, *A*. *clavatus*, *A*. *niger*, *A*. *oryzae*, *A*. *flavus*, and *A*. *nidulans*. RgsA of *A. fumigatus* shows 71.1% to 97.0% identity and 80.2% to 98.1% similarity to the RgsA-like proteins of other aspergilli.

### 2.2. RgsA Attenuates Hyphal Growth and Asexual Development

To characterize the function of *rgsA* in asexual development, we used the DJ-PCR method to generate the Δ*rgsA* mutant by replacing the ORF with the *A. nidulans pyrG* + marker [21]. As shown in Figure 1A–D, the radial growth of the Δ*rgsA* mutant was significantly higher than that of WT and complemented (C′) strains. Conidia per growth area further indicated that conidia production in the Δ*rgsA* mutant (1.47 × 10^8^ conidia/cm^2^) was increased (*p* < 0.01) to about 160% of the WT strain (Figure 1D). Even in the stationary culture, asexual sporulation of the mutant was rapid and increased in comparison with that of WT and C′ strains (Figure 1B). We then performed quantitative RT-PCR (qRT-PCR) for examination of the mRNA levels of the key asexual developmental regulators *brlA*, *abaA*, and *wetA* in WT and Δ*rgsA* mutant strains. As shown in Figure 1E, mRNA levels were significantly higher in the Δ*rgsA* mutant than that of WT, especially at 24 h post developmental induction. These data suggest that RgsA is needed for adequate control of growth and development in *A. fumigatus*.

### 2.3. RgsA Down-Regulates a cAMP-Dependent Protein Kinase A Signaling Pathway

A previous study demonstrated that the absence of *rgsA* resulted in elevated conidial germination in *A. nidulans*, with or without an external carbon source [20]. To investigate a potential role of RgsA in controlling spore germination, we monitored the kinetics of germ tube emergence in the Δ*rgsA* mutant in comparison to that of WT every 2 h after inoculation. As shown in Figure 2A, conidia of WT and Δ*rgsA* strains began to germinate after 6 h of incubation and showed no significant differences in the presence of glucose. To test further, conidia of WT and Δ*rgsA* strains were inoculated in a no carbon source medium and observed for germination. Both strains started form germ tubes, about 2% and 3%, respectively at 6 h. At 12 h of incubation, while only about 10% of WT conidia were germinated, 35% of the Δ*rgsA* conidia germinated (Figure 2A), indicating that RgsA may negatively regulate conidial germination, potentially sensing the external carbon source. To investigate the relationship between RgsA and a cAMP-PKA signaling pathway, we assessed PKA’s activity using a peptide substrate, kemptide. While all the strains we tested exhibited similar levels of PKA activity in the presence of cAMP, the Δ*rgsA* mutant showed PKA activity even in the absence of cAMP (Figure 2B). To investigate this further, we analyzed mRNA levels of PKA pathway related genes. As shown in Figure 2C, *acyA,* and *pkaC1* mRNA levels were significantly higher in the Δ*rgsA* mutant than in WT strain. These results indicate that RgsA is required for properly controlling the expressions of *acyA* and *pkaC1*, and may negatively regulate a cAMP-PKA signaling pathway.

### 2.4. Differential Roles of RgsA in Oxidative Stress Response

To test the potential role of RgsA in the oxidative stress response, we incubated WT, Δ*rgsA*, and C′ strains on solid MMY containing paraquat (PQ) or H_2_O_2_. As shown in Figure 3A, the Δ*rgsA* mutant exhibited a significantly increased tolerance to PQ and H_2_O_2_ (Figure 3B). To further dissect RgsA’s function, we analyzed activities of catalase and SOD. In *A*. *fumigatus*, superoxide dismutases (SODs) encode four genes and Sod1p and Sod2p are concerned with the detoxification of the intracellular superoxide radical [22]. Sod1p is a cytoplasmic, cyanide-sensitive Cu/ZnSOD and Sod2p is a mitochondrial, cyanide-insensitive MnSOD [22]. The activity of Sod1p was slightly reduced in the Δ*rgsA* mutant, but Sod2p’s activity was higher than that of WT strain (Figure 3C), suggesting that the increased resistance of the Δ*rgsA* mutant to PQ might be associated with higher Sod2p activities. While the activity of conidia-specific catalase (CatAp) was increased about 1.7-fold, mycelia-specific catalase (Cat1p) was decreased about 1.5-fold in the Δ*rgsA* mutant compared to the WT and C′ strains (Figure 3D). In addition, mRNA levels of genes encoding stress-activated signaling pathway elements were increased about 1.2 to 2.6-fold by Δ*rgsA* (Figure 3E). Taken together, these results suggest that RgsA is needed for proper regulation of responses to external oxidative stresses.

### 2.5. Transcriptome Analysis: Differentially Expressed Gene (DEG) Analysis and Functional Classification

To obtain the genome-wide expression changes resulting from the absence of *rgsA*, we carried out Quant-Seq (3′mRNA-Seq) using Δ*rgsA* and WT strains. Two strains showed a high level of correlation (R^2^ = 0.9095; Figure 4A). Of the 25,202,373 reads, 8277 genes exhibited slight fold changes (–1 < log_2_FC < 1) (Figure 4B) and 440 genes showed a significant, differential expression (at least 2.0 fold, *p* < 0.05), of which 229 were up-regulated (Appendix A) and 211 were down-regulated (Appendix A). In molecular function gene ontology (GO) categories, “oxidoreductase activity” and “cofactor binding activity” were up-regulated, whereas “transporter activity” was down-regulated. The top, significant biological process GO categories were “pathogenesis” (Figure 4C). Most of the up-regulated genes were predicted to encode, in the numerous genes of the putative GT biosynthetic cluster in *A. fumigatus*: MFS GT efflux transporter GliA (AFUA_6G09710), cytochrome P450 oxidoreductase GliC (AFUA_6G09670), GliF (AFUA_6G09730), glutathione S-transferase GliG (AFUA_6G09690), aminotransferase GliI (AFUA_6G09640), membrane dipeptidase GliJ (AFUA_6G09650), O-methyltransferase GliM (AFUA_6G09680), methyltransferase GliN (AFUA_6G09720), nonribosomal peptide synthase GliP (AFUA_6G09660), or thioredoxin reductase GliT (AFUA_6G09740) (Appendix A). The majority of the down-regulated genes were predicted to encode glucanases, including endoglucanase (AFUA_3G03950, AFUA_4G07850, AFUA_7G06740, and AFUA_6G116000), 1,4-β-D-glucan-cellobiohydrolase (AFUA_6G11610), and exo-β-1,3-glucanase (AFUA_6G13270) (Appendix A).

### 2.6. RgsA Down-regulates GT Production and Virulence

Gliotoxin is an important determinant of the virulence of *A. fumigatus* [23]. Among the putative GT biosynthetic 13-gene cluster in the genome of *A. fumigatus* [24,25], *gliA*, *gliC*, *gliF*, *gliG*, *gliI*, *gliJ*, *gliM*, *gliN*, *gliP*, and *gliT* have been identified as the up-regulated genes with the maximum fold changes in the mutant relative to WT. These findings led us to test a role for the RgsA in GT production per se. We examined levels of GT in WT and Δ*rgsA* strains by TLC, and found that the Δ*rgsA* mutant produced a significantly higher amount of GT than WT and C′ strains (Figure 5A). Next, we tested the effects of RgsA on virulence using the outbred CD1 non-neutropenic mice that were generated by subcutaneous injection of cortisone acetate (10 mg/mouse). Conidia (1 × 10^7^) of WT, Δ*rgsA*, and C′ strains were inoculated to the mouse, and the survival was recorded every 12 h. The most rapid lethality appeared with the Δ*rgsA* mutant and this strain was significantly more virulent than the other strains (*p* = 0.0006). As shown in Figure 5B, mice inoculated with conidia of either WT or C′ strains began to die at day 2.5 post-inoculation, with the survival rate continuing to decrease over the course of the experiment. Five days after infection, about 50% of mice in both strains died. However, the mortality rates caused by Δ*rgsA* strain were higher than those of WT and C′ strains: about 90% of mice had died in 4.5 days and all mice had died by 7.5 days after infection. Furthermore, the loss of *rgsA* significantly enhanced the pulmonary fungal burden of mice (Figure 5C). To better understand the fate of *A. fumigatus* inoculated into mouse, the lung tissues of infected mice were fixed with formalin 72 h post-infection and sectioned for histopathology. Figure 5D shows Hematoxylin and Eosin (H&E) and periodic acid-Schiff (PAS) stained sections of infected mice with three strains. Histopathological examination of lung tissues from WT, Δ*rgsA*, and C′ strains of infected mice demonstrated that different mortality rates among the groups were associated with different pulmonary fungal burdens. While WT-strain infected mice revealed marked increases of neutrophil recruitment within the bronchi and the peribronchial region; mild necrosis; and the infiltration of numerous red blood cells into the parenchyma, Δ*rgsA* strain infected mice showed bronchial wall disruption, severe necrosis, scattered regional inflammation, and mild hemorrhaging in lung parenchyma (Figure 5D). Next, to investigate whether more severe lung damage observed in mice infected with Δ*rgsA* could be attributed to the extent of fungal invasion, lung sections from three different groups were stained with PAS. WT-strain infected mice revealed the presence of the most conidia within bronchi and the peribronchial region, and rare hyphae growth was detectable. In contrast, lung sections from Δ*rgsA*-strain infected mice displayed numerous spore germinations and aggressive invasion of hyphae into parenchyma (Figure 5D). Of note, C′ strain infected mice showed similar phenotypes to those of WT-strain infected mice, suggesting that RgsA is a negative regulator of fungal virulence.

### 2.7. RgsA Plays a Positive Role in Endoglucanase Activity

In RNA-seq analysis, we found that genes encoding numerous carbohydrate-metabolizing enzymes were significantly down-regulated (Log_2_FC = –2.07 to –3.96) in the Δ*rgsA* strain compared to WT (Appendix A; Figure 6A). Furthermore, most of those genes are predicted to encode cellulolytic enzymes, including cellobiohydrolase (AFUA_6G07070, AFUA_6G11610, and AFUA_3G01910), endo-1,4-β-glucanase (AFUA_3G03870 and AFUA_1G05290), β-glucosidase (AFUA_1G14710 and AFUA_1G05770), and cellobiose dehydrogenase (AFUA_2G17620). To test whether RgsA affects endoglucanase activity, we determined the enzyme activity by culturing three strains in liquid MMY containing 0.1% β-glucan. As shown in Figure 6B, both β-1,4 carboxymethyl cellulose (CMC) and β-1,3 (laminarin) endoglucanase activities were significantly reduced irrespective of location, except for the intracellular β-1,4-endoglucanase activity. Those results suggest that RgsA plays a positive role in the degradation of glucan and may play an important roles in saprophytic nutrition.

## 3. Discussion

G-protein signaling is conserved in all eukaryotes, playing a key role in sensing and transmitting signals into cells to lead to appropriate responses [26]. RGS proteins constitute a diverse and multifunctional family of proteins that play an important role in controlling heterotrimeric G protein signaling [9,27]. In *A. nidulans*, the homologue of *rgsA* was identified as a specific RGS protein that negatively regulates GanB function. This RGS-Gα pair plays an important role in upstream regulation of asexual sporulation, germination, and stress responses to environmental changes [20]. In this study, the deletion of *rgsA* resulted in increased radial growth and conidiation and elevated mRNA levels of central asexual development regulators *abaA*, *brlA*, and *wetA* compared to the WT strain (Figure 1). Based on all that, we can propose that RgsA is crucial for the proper control of growth and asexual development in *A*. *fumigatus*.

Previous studies described that *A. nidulans* RgsA negatively regulates GanB (G protein α subunit) and that the RgsA–GanB pair governs upstream regulation of spore germination and carbon source sensing, in part via the cAMP-PKA pathway [11,20,28]. It has been shown that activation of cAMP-PKA pathway in *A. nidulans* can induce germination by various carbon sources. In *A. fumigatus*, the Δ*rgsA* mutant and WT began to germinate after 4 h of incubation and the Δ*rgsA* mutant showed higher germination rates than those of WT at 8 h. After 12 h of incubation, both the Δ*rgsA* mutant and WT strains exhibited nearly 100% germination. In addition, we found that the germination rates of the Δ*rgsA* mutant conidia were significantly higher than those of WT strain in the absence of a carbon source. Furthermore, the Δ*rgsA* strain showed higher PKA activity in the absence of cAMP and mRNA levels of PKA signaling components. Collectively, these results imply that RgsA serves as an important negative, likely upstream regulator of the cAMP-PKA signaling pathway (Figure 2).

The ability to sense external stress and respond appropriately is vital for the survival of fungal cells in stress conditions, and is a key factor for *A. fumigatus* to establish successful invasive aspergillosis. *A. fumigatus* is exposed to variety of stressors in both the environment and the host. During infection, *A. fumigatus* encounters various microenvironments, which force the fungus to grow under different stress conditions during pathogenesis [29]. These stress conditions include temperature, oxidative stress, pH changes, and the low availability of macro and micro-nutrients [1,3,30,31,32,33]. Our results show that RgsA is involved in the response to oxidative stress, since the *rgsA* deletion mutant was highly resistant to PQ and H_2_O_2_ (Figure 3).

The human pathogen *A. fumigatus* produced secondary metabolites such as GT, fumagillin, fumitremorgin, gibberellin, and helvolic acid a [34]. GT is a major and the most potent toxin of this fungus, and a member of the epipolythiodioxopiperazine class of toxins. Transcriptome analysis revealed that 440 genes were significant differentially expressed by the loss of *rgsA* (at least 2.0-fold, *p*-value < 0.05) with 229 up-regulated and 211 downregulated genes (Figure 4). Interestingly, most of the up-regulated genes in the Δ*rgsA* mutant were predicted to encode genes in the GT biosynthesis gene cluster (Appendix A). GliP is the first step in biosynthesis of gliotoxin (GT) and is essential for GT production and the absence of this gene resulted in GT deficient strains [23,35,36,37]. GliA is a member of the major facilitator superfamily of transporters. The GT oxidoreductase GliT functions in the protection of the fungus against external GT and is indispensable for GT biosynthesis [24,38]. The *gliI* gene is predicted to encode aminotransferase and represents the first gene of a functionally proven C–S lyase involved in a secondary metabolite pathway [39]. Glutathione S-transferase GliG is not involved in the self-protection of *A. fumigatus* against exogenous gliotoxin [40]. The *gliM* and *gliN* genes are predicted to encode an *O*-methyltransferase and methyltransferase respectively [37]. GliJ, a putative dipeptidase, shows higher expression in biofilms and *gliC* is the predicted cytochrome P450 monooxygenase and acts upstream of *gliG* in the gliotoxin biosynthesis pathway [39]. The Δ*rgsA* mutant produced a significantly higher amount of GT than WT and C′ strains, and caused higher the mortality rates in mice. The lung sections of the Δ*rgsA*-strain infected mice displayed numerous spore germinations and the aggressive invasion of hyphae into parenchyma (Figure 5), suggesting that RgsA down-regulates GT production and fungal virulence.

*A. fumigatus* has a saprophytic lifestyle in decaying organic and plant material, and this fungus has a numerous genes that encode a wide range of hydrolases with the capacity to degrade the major plant cell wall polymers [41]. The majority of down-regulated genes, by the loss of *rgsA*, are carbohydrate-metabolism-related genes (Appendix A; Figure 6A). The cellobiohydrolase *celD* involved in cellulose breakdown and its predicted gene pair with *cbhB* (AFUA_6G11610) are similar to *cbhA* and *cbhB* in *A. nidulans* (AN5176 and AN0494), which were induced by cellulose and repressed by glucose [42]. The endoglucanase activity of the mutant was lower than WT and C′ strains in β-glucan containing medium (Figure 6B), indicating that RgsA is involved in the degradation of β-glucan and may play an important role in saprophytic nutrition.

In summary, we propose a genetic model depicting the regulatory roles of RgsA in *A. fumigatus* (Figure 7). In the absence of *rgsA*, the cAMP-PKA and stress-activated signaling pathway may be activated, which might lead to increased GT production, virulence, and tolerance to environmental stresses, whereas polysaccharide breakdown activity may be inhibited by the loss of *rgsA*. Further studies are needed to identify the targeted G protein α subunit(s) of RgsA and the potential downstream signaling pathways governing the stress responses and virulence of *A. fumigatus*.

## 4. Materials and Methods

### 4.1. Fungal Strains and Culture Conditions

*A. fumigatus* AF293.1 (*AfpyrG1*) was used to generate the Δ*rgsA* mutant and AF293 was used as a wild type (WT). Fungal strains were grown on glucose minimal medium (MMG) or MMG with 0.1% yeast extract (MMY) with appropriate supplements, as described previously [43]. For liquid-submerged culture, about 5 × 10^5^ conidia/mL were inoculated into liquid MMY and incubated at 37 °C, 250 rpm. For phenotypic analyses of respective strains on air-exposed culture, conidia (1 × 10^5^) of relevant strains were spotted on solid medium and incubated at 37 °C for 4 days.

### 4.2. Generation of the rgsA Null Mutant

The double-joint PCR method was used to generate the *rgsA* deletion mutant [21] and the oligonucleotides used in this study are listed in Appendix A. Briefly, using WT (AF293) genomic DNA as a template, 5′ and 3′-flaking regions of the *rgsA* ORF were amplified with the primer pairs of Oligo926:Oligo927 and Oligo928:Oligo929, respectively. The selective marker *A. nidulans pyrG* was amplified with Oligo697 and 698 using *A. nidulans* FGSC A4’s genomic DNA as a template. The three amplicons were fused and the second round of PCR was performed. Using the second round PCR product as a template, the deletion construct was amplified with nested primer pairs Oligo930 and Oligo931, and the resulting PCR amplicon was introduced into AF293.1 via transformation. The *rgsA* deletion mutant colonies were isolated and confirmed by PCR followed by *SmaI* digestion (Appendix A). To complement the *rgsA* null mutant, the single joint PCR (SJ-PCR) method was used [21]. The ORF of *rgsA* with a promoter and a terminator was amplified with primer pairs; the 3′ reverse primer carried overlapping sequences with the *ptrA* gene’s 5′ end. Amplification of the *ptrA* gene was carried out with primer pairs; the 5′ forward primer carried overlapping sequences with *ptrA* gene’s 3′ end. The final amplicon was amplified with the nested primer pair oligo 930/oligo 931 and introduced into a Δ*rgsA* strain (Appendix A).

### 4.3. Nucleic Acid Isolation and Manipulation

Total RNA was isolated using Trizol reagent (Invitrogen, USA). RNA quality was assessed by Agilent 2100 bioanalyzer using the RNA 6000 Nano Chip (Agilent Technologies, Amstelveen, The Netherlands), and RNA quantification was performed using ND-2000 Spectrophotometer (Thermo Inc., USA). Levels of mRNA of select genes were analyzed with appropriate oligonucleotide pairs (Appendix A). For RNA-seq analyses, 3 day-old cultures of WT and Δ*rgsA* strains were harvested from solid MMY. Quantitative RT-PCR assays were performed as previously described [13,44]. Briefly, conidia (5 × 10^5^ conidia/mL) of three strains were inoculated into liquid MMY with appropriate supplements and incubated at 37 °C, 250 rpm. Individual mycelial samples, which were collected at designated time points, were homogenized using a Mini Bead beater in the presence of 1 mL of TRIzol^®^ reagent (Invitrogen, USA) and 0.3 mL of silica/zirconium beads (BioSpec Products, USA). QRT-PCRs were performed according to the manufacturer’s instructions using a Rotor-Gene Q (Qiagen, USA). Each run was assayed in triplicate in a total volume of 20 µL containing the RNA template, One Step RT-PCR SYBR Mix (Doctor Protein, Korea), reverse transcriptase, and 10 pmole of each primer (Appendix A). Reverse transcription was carried out at 42 °C for 30 min. PCR conditions were 95 °C/5 min for one cycle, followed by 95 °C and 55° C/30 s for 40 cycles. Amplification of one specific target DNA was checked by melting curve analysis. The expression ratios were normalized to *ef1α* expression and calculated according to the ΔΔCt method [44].

### 4.4. Phenotypic Analyses

Germination rates were determined with a previously-established method with some modifications [45]. The conidia of WT and Δ*rgsA* were inoculated in 5 mL of MMY, or media without a carbon source, and incubated at 37 °C. Levels of both isotopic growth and germ tubing were examined every 2 h after inoculation under a microscope. To assess the role of RgsA in stress responses, WT and Δ*rgsA* strains were grown on various media, the base being MMG as described above. To test for oxidative stress, PQ (200 µM) and hydrogen peroxide (2 mM) were added to MMG after autoclaving. All experiments were performed in triplicate. To determine the production of gliotoxin (GT), 1 × 10^7^ conidia/mL of each strain were inoculated into 5 mL complete medium in a test tube and incubated at 37 °C slanted for 7 days. GT was extracted with chloroform as described previously [46]. Individual chloroform extracts were air-dried and resuspended in 100 µL of methanol. Aliquots (10 μL) of each sample were applied to a TLC silica plate containing a fluorescence indicator (Kiesel gel 60, E. Merck, USA). The TLC plate was developed with chloroform:methanol (9:1, *v*/*v*).

### 4.5. Enzyme Assay

To measure the PKA activity, the Non-Radioactive cAMP-Dependent Protein Kinase Assay kit (Promega, USA) was used. Homogenized mycelia of each strain were suspended in extraction buffer [47] and incubated in ice for 15 min. After centrifugation, 10 µL of supernatant (3 mg/mL protein) was used for determining PKA activity. To determine catalase and SOD activities, the conidia (1 × 10^5^) of relevant strains were inoculated into liquid MMY with appropriate supplements and incubated at 37 °C, 250 rpm, for 24 h. Then, oxidative stress agents (200 µM PQ or 2 mM H_2_O_2_) were added, and the conidia were further incubated for 24 h. The mycelia were disrupted with glass beads in 20 mM pH 7.5 phosphate buffer supplemented with a protease inhibitor cocktail. Protein contents were quantified with the Bradford reagent (Bio-Rad, USA), using bovine serum albumin as the standard. Catalase activity was visualized by negative stain with ferricyanide [48] and SOD activity was detected as inhibition of the reduction of NBT [49]. For testing endoglucanase activity, fungal strains were grown on MMY with 1% barley β-glucan which contained both β-1,3 and β-1,4-glucan. The enzyme activities of culture filtrates and mycelial extracts were measured by DNS method [50] using CMC (for β-1,4-glucanase) and laminarin (for β-1,3-glucanase) as the substrates. One unit of endoglucanase activity was defined as the amount of enzyme that liberated reducing sugars equivalent to 1 μmol of glucose under the assay conditions.

### 4.6. Murine Virulence Assay

For the immunocompromised mouse model, we used outbred CrlOri:CD1 (ICR) (Orient Bio Inc, Korea) female mice (30 g in body weight, 6 to 8 weeks old), which were housed five per cage and had access to food and water ad libitum. Mice were immunosuppressed with subcutaneous injections of cortisone acetate (Sigma-Aldrich, USA) at 10 mg/mouse for 4 days prior to infection and cortisone acetate injected subcutaneously, 10 mg/mouse, 2 days prior to infection. At days 0, 3, and 6 post-infection, administrations were repeated with cortisone acetate (10 mg/mouse). For conidia inoculation, mice were anesthetized with isoflurane, and then intranasally infected with 1 X 10^7^ conidia of *A*. *fumigatus* strains (10 mice per fungal strain) in 30 µL of 0.01% Tween 80 in PBS. Mice were monitored every 12 h for survival for 8 days after the challenge. Mock mice included in all experiments were inoculated with sterile 0.01% Tween 80 in PBS. Mice were checked every 12 h for survival and Kaplan–Meier survival curves were analyzed using the log-rank (Mantel–Cox) test for significance (*p* < 0.01).

### 4.7. Fungal Burden in Mice

Fungal burdens were obtained by the qPCR approach from a previous study [51] with *A*. *fumigatus*-specific primers [52]. Briefly, a complete lung from each mouse was lyophilized and ground to a fine powder. Approximately 50 mg of each powdered lung was used for DNA extraction. The amounts of DNA from the lung tissues were quantified by spectrophotometer and all samples were diluted to 100 ng/μL. Standard *C_T_*-versus-concentration plots were constructed using *A*. *fumigatus* DNA (20 to 0.2 ng per reaction, R = 0.992) and the concentration of fungal DNA in each sample was calculated.

### 4.8. Ethics Statement

All of the animal procedures in this study were reviewed and approved by the Institutional Animal Care and Use Committee of Daejeon University (DJUARB2019-024).

### 4.9. RNA-seq Experiment and Analyses

Construction of the RNA-seq library was performed using QuantSeq 3′ mRNA-Seq Library Prep Kit (Lexogen, Inc., Austria) according to the manufacturer’s instructions. In brief, each 500 ng total RNA was prepared and an oligo-dT primer containing an Illumina-compatible sequence at its 5′ end was hybridized to the RNA, and reverse transcription was performed. After degradation of the RNA template, second strand synthesis was done by a random primer containing an Illumina-compatible linker sequence at its 5′ end. The double-stranded library was purified by using magnetic beads and amplified to add the complete adapter sequences required for cluster generation. The finished library was purified from PCR components. High-throughput sequencing was performed as single-end 75 sequencing using NextSeq 500 (Illumina, Inc., San Diego, CA, USA). QuantSeq 3′ mRNA-Seq reads were aligned using Bowtie2 [53]. Bowtie2 indices were either generated from genome assembly sequences or the representative transcript sequences for aligning the genome and transcriptome. The alignment file was used to assemble transcripts, estimating their abundances and detecting the differential expressions of genes. Differentially-expressed genes were determined based on counts from unique and multiple alignments using coverage in Bedtools [54]. The RT (read count) data were processed based on Quantile normalization method using EdgeR within R using Bioconductor [55]. Gene classification was based on searches done by DAVID (http://david.abcc.ncifcrf.gov/) and Medline databases (http://www.ncbi.nlm.nih.gov/).

### 4.10. Data Analysis

Comparisons among groups of two and larger groups were performed by the unpaired Student’s *t*-test and one-way ANOVA, respectively. GraphPad Prism 4 (GraphPad Software, Inc., San Diego, CA, USA) was used for the statistical analyses and graphical presentation of survival curve. *p* < 0.05 was considered to be statistically significant.

### 4.11. Data Availability

The RNA-seq data are available from NCBI Gene Expression Omnibus (GEO) database (the accession number is GSE123744).

## 5. Conclusions

Canonical regulator of G protein signaling (RGS) proteins accelerate the hydrolysis of GTP bound to Gα, leading the formation of the inactive hetero-trimer and turning off G protein signaling [9]. A better understanding of individual RGS proteins in *A. fumigatus* may lead to the identification of novel antifungal drug targets and the control of human-pathogenic fungi.

Our study identified that RgsA can negatively regulate the cAMP-PKA and stress-activated kinase signaling pathways, which leads to a disturbed lifecycle; i.e., enhanced asexual development, increased GT production, increased virulence, and a higher oxidative stress response. However, the precise mechanisms of RgsA-mediated regulation of hetero-trimeric G protein signaling and downstream signaling pathways are not understand yet. To fully understand the function of RgsA in *A. fumigatus*, protein–protein interactions between RgsA and G proteins can be studied to identify the targets, and the outcomes of those studies may provide an expanded model for understanding G protein signaling in other human-pathogenic fungi.

## Figures and Tables

**Figure 1 ijms-20-05628-f001:**
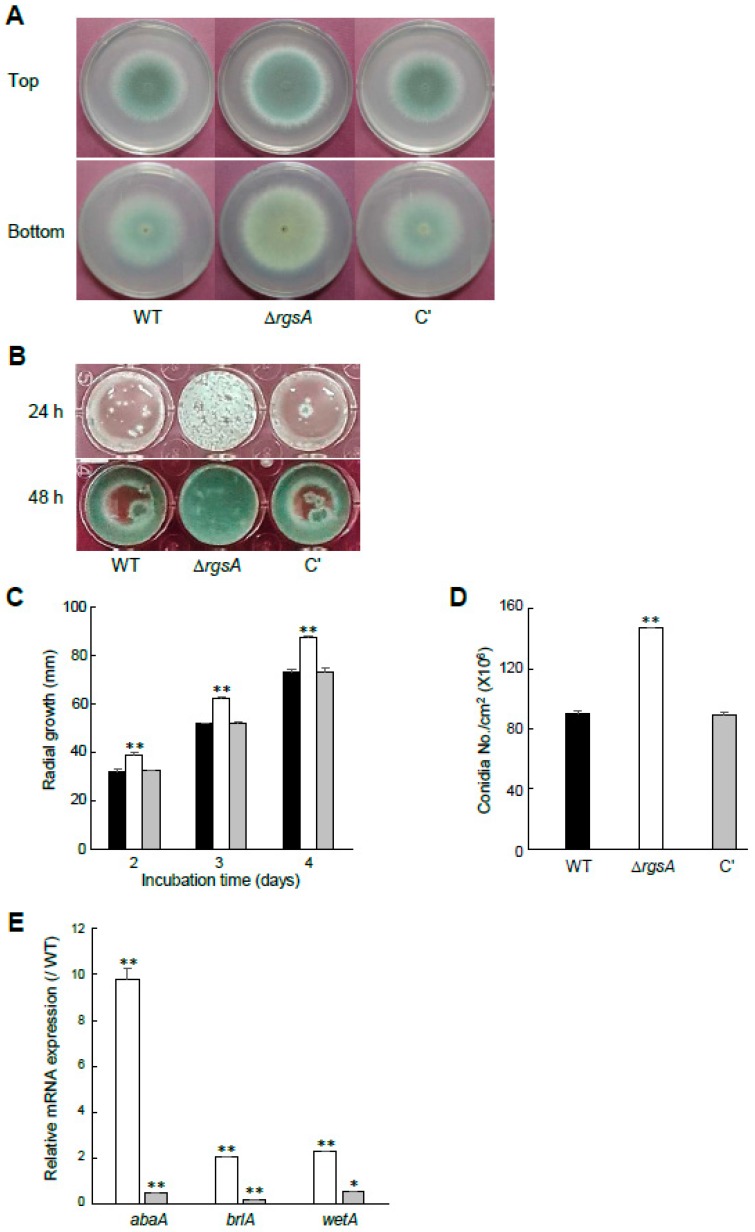
The role of RgsA in fungal growth and development. (**A**) Colony photographs of WT (AF293), Δ*rgsA*, and complemented (C′) strains point-inoculated on solid MMY and grown for 3 days. (**B**) Growth of three strains on liquid MMY stationary culture at indicated time. (**C**) Colony diameters of WT (black bar), Δ*rgsA* (white bar), and C′ (gray bar) strains. (**D**) Conidia numbers produced by each strain per growth area. (**E**) mRNA levels of the key asexual developmental regulators in the Δ*rgsA* mutant relative to WT at 24 (white bar) and 48 h (gray bar) determined by qRT-PCR. Fungal cultures were grown in liquid MMY and mRNA levels were normalized using the *ef1α* gene. Data are presented as the means ± standard deviations from three independent experiments. ANOVA test: ** *p* < 0.01.

**Figure 2 ijms-20-05628-f002:**
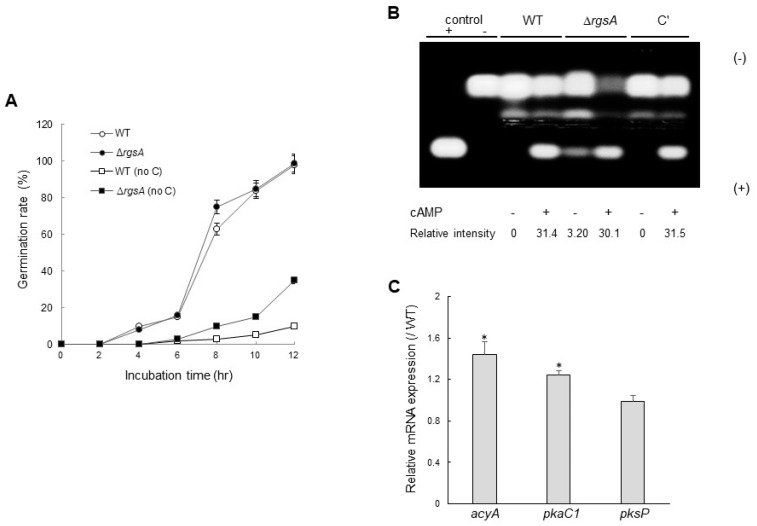
The role of RgsA in spore germination and the cAMP-PKA signaling pathway. (**A**) Kinetics of germ tube outgrowth of WT and Δ*rgsA* strains inoculated in liquid MMG at 37 °C in the presence or absence of glucose. (**B**) PKA activity of WT (AF293), Δ*rgsA*, and complemented (C′) strains, as monitored by gel electrophoresis. A phosphorylated substrate migrates toward the + anode. Each strain was grown in MMG for 3 days at 37 °C and mycelial extract was analyzed. (**C**) Levels of *acyA*, *pkaC1*, and *pksP* mRNA in three strains analyzed by qRT-PCR. * *p* < 0.05.

**Figure 3 ijms-20-05628-f003:**
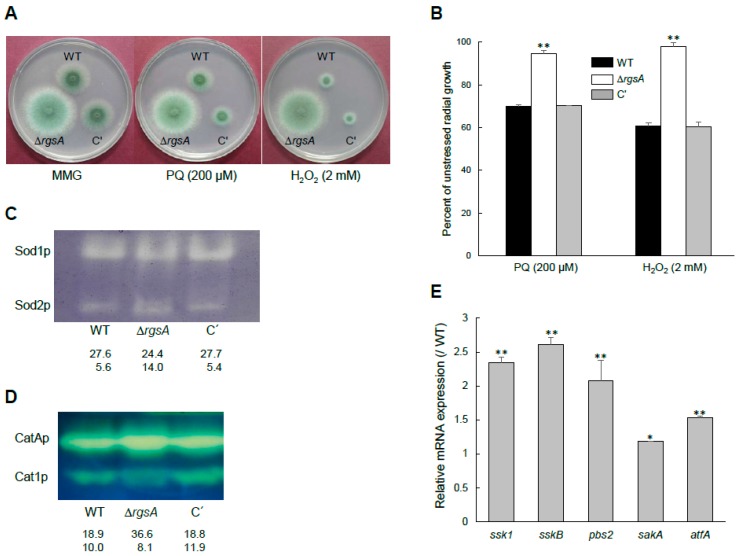
RgsA’s functions in the oxidative stress response. (**A**) Colony appearance after point inoculation of 1 X 10^5^ conidia on solid MMG with oxidative stressors. (**B**) Quantification of growth inhibition was measured by colony diameters after 48 h of incubation at 37 °C. (**C**) SOD activity in WT, Δ*rgsA*, and C′ strains shown in non-denaturing polyacrylamide gels. (**D**) Catalase activity in WT, Δ*rgsA*, and C′ strains. Relative intensities of each enzyme are shown below. (**E**) Levels of stress-activated genes’ mRNA in three strains analyzed by qRT-PCR. Statistical significance was determined by ANOVA and a Student *t*-test: ** *p* < 0.01; * *p* < 0.05.

**Figure 4 ijms-20-05628-f004:**
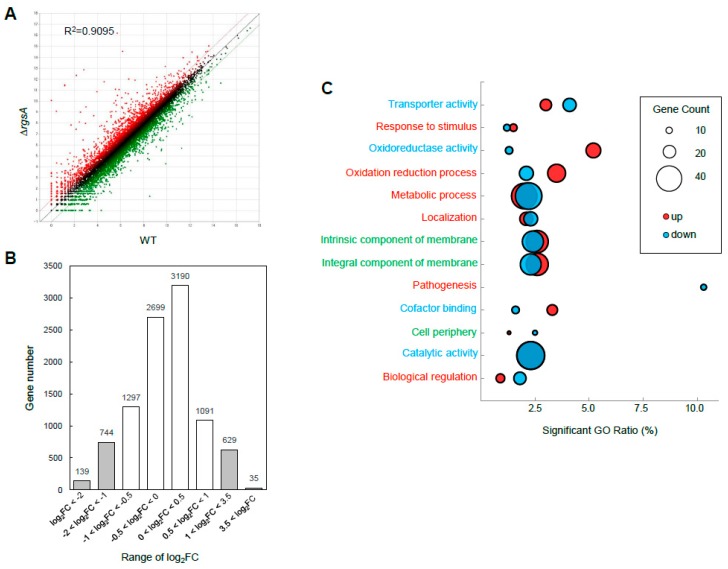
Genome-wide expression analysis between WT and Δ*rgsA* strains. (**A**) Linear fitted model showing the correlation between overall gene expression for WT and Δ*rgsA* strains. The correlation coefficient R^2^ is indicated. (**B**) Histograms showing general transcriptomic results and columns in white fall in the −1 < log_2_FC < 1 fragment count range with low differential expression values. (**C**) Functional categories of DEGs. The red and blue circles represent genes whose mRNA levels were increased and decreased in the Δ*rgsA* mutant, respectively.

**Figure 5 ijms-20-05628-f005:**
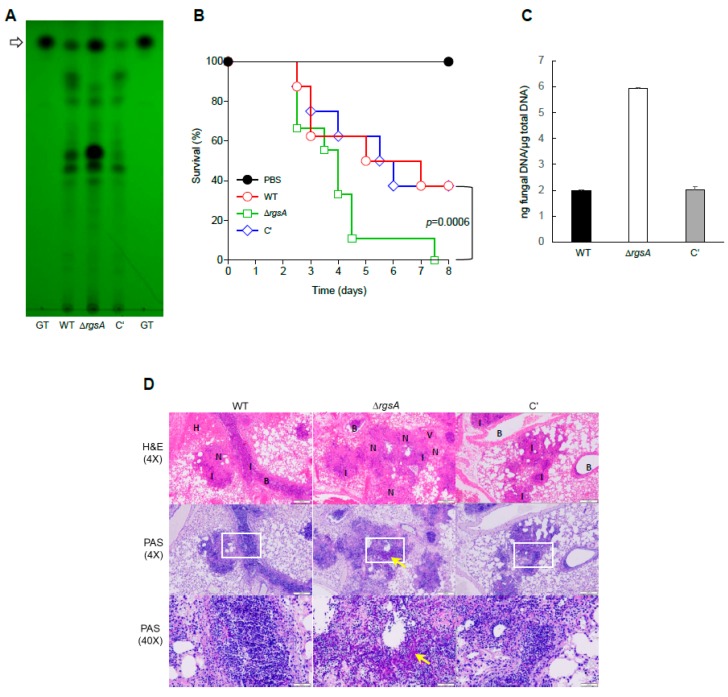
The role of RgsA in GT production and virulence. (**A**) Determination of GT production in WT, Δ*rgsA*, and C′ strains. The culture supernatant of each strain was extracted with chloroform and subjected to TLC. The arrow indicates GT. (**B**) Survival curve of mice infected with WT, Δ*rgsA*, and C′ strains (n = 10/group). (**C**) Fungal burden in the lung tissue by RT-PCR method. (**D**) Lung sections from infected WT, Δ*rgsA*, and C′ strains were stained with Hematoxylin and Eosin (H&E) or periodic-acid Schiff (PAS). Yellow arrow indicates highly-dense fungal hyphae visualized by magenta color. H: hemorrhage; B: bronchiole; I: inflammation; N: necrosis; V: blood vessel. Scale bars = 200 µm (4×) and 50 µm (40×).

**Figure 6 ijms-20-05628-f006:**
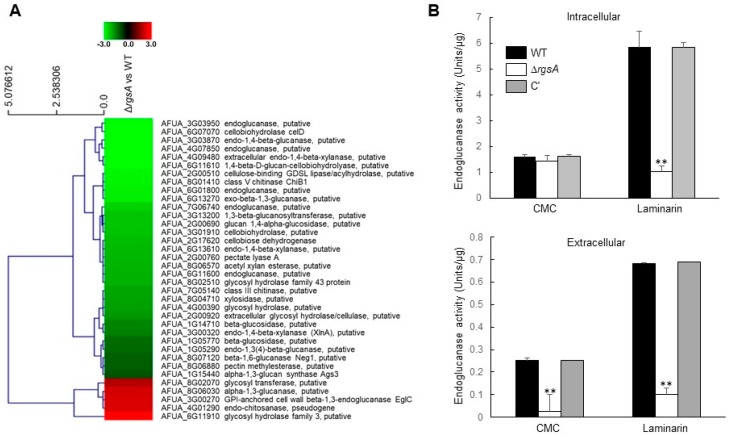
RgsA is needed for proper carbohydrate metabolism. (**A**) Heat map of those genes encoding carbohydrate-metabolism related enzymes. Most of genes were down-regulated by the lack of *rgsA*. (**B**) Intracellular and extracellular endoglucanase activities of three strains. ANOVA test: ** *p* < 0.01.

**Figure 7 ijms-20-05628-f007:**
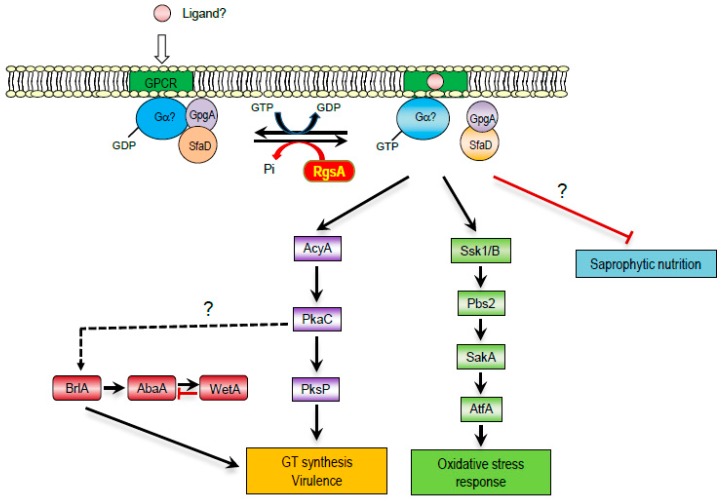
A model depicting the predicted roles of RgsA. RgsA negatively regulates the cAMP-PKA signaling pathway and stress-activated signaling pathway. This is predicted to be accomplished by RgsA’s role in attenuating the target heterotrimeric G proteins. The lack of RgsA leads to the accelerated and prolonged activation of the target G proteins; thus, enhanced activation of the cAMP-PKA signaling pathway and stress-activated signaling pathway, which in turn leads to increased production of GT, enhanced virulence, and an elevated oxidative stress response. In contrast, polysaccharide breakdown activity may be inhibited by the lack of *rgsA*.

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
