# Peer review of "RgsA Attenuates the PKA Signaling, Stress Response, and Virulence in the Human Opportunistic Pathogen Aspergillus fumigatus"

_ijms, 2019, doi:10.3390/ijms20225628_

Round 1

Reviewer 1 Report

Lwin et al have performed an interesting study to characterized the RgsA gene and some of its functions related to the human opportunistic pathogenic fungus Aspergillus fumigatus. The authors conclude that  RgsA plays a multiple role in governing growth, development, stress response, virulence, and external polymer degradation, especially attenuates PKA signaling, stress response and virulence in this fungus infection.

The study is very interesting and adds new information to the literature. Some comments.

I feel that the abstract is a bit confusing and unstructured. Please simplify and structure the main messages (introduction, main objectve, results, methods, working hypothesis and conclusions) to improve understanding of the reader. The methods are very well explained, and the tables/graphics are very appropriate I miss some sentences on the potential limitation of the study (discussion section) I also miss some words about the translational value of the findingd of this study. How this study could help physician in a AFumigatus infection?. What are the future clinical implications of these findings?. Please explain A section of future and new challenge on this topic would be desirable.

Author Response

Comments and Suggestions for Authors

Lwin et al have performed an interesting study to characterized the RgsA gene and some of its functions related to the human opportunistic pathogenic fungus Aspergillus fumigatus. The authors conclude that  RgsA plays a multiple role in governing growth, development, stress response, virulence, and external polymer degradation, especially attenuates PKA signaling, stress response and virulence in this fungus infection.

The study is very interesting and adds new information to the literature. Some comments.

I feel that the abstract is a bit confusing and unstructured. Please simplify and structure the main messages (introduction, main objectve, results, methods, working hypothesis and conclusions) to improve understanding of the reader.

⇒ We appreciate this comment and have revised Abstract to improve the clarity and structure.  

The methods are very well explained, and the tables/graphics are very appropriate I miss some sentences on the potential limitation of the study (discussion section)

⇒ We added a sentence at the end of Discussion (line 292-294)

“Further studies are needed to identify the target G protein alpha subunit(s) of RgsA and the potential downstream signaling pathways governing the stress responses and virulence in A. fumigatus.”

I also miss some words about the translational value of the findingd of this study. How this study could help physician in a AFumigatus infection?. What are the future clinical implications of these findings?. Please explain A section of future and new challenge on this topic would be desirable.

⇒ Thanks again! We added the section 5. “Conclusions” and included your suggestions (line 426-438).

“Canonical regulator of G protein signaling (RGS) proteins accelerate the hydrolysis of GTP bound to Gα, leading the formation of the inactive hetero-trimer and turning off G protein signaling [9]. A better understanding of individual RGS proteins in A. fumigatus may lead to identification of novel antifungal drug targets and control of human pathogenic fungi.

Our study identified that RgsA can negatively regulate cAMP-PKA and stress activated kinase signaling pathway, which leads to disturbed lifecycle, i.e., enhanced asexual development, increased GT production, virulence, and oxidative stress response. However, the precise mechanisms of RgsA-mediated regulation of hetero-trimeric G protein signaling and downstream signaling pathways are not understand yet. To fully understand the function of RgsA in A. fumigatus, protein-protein interactions between RgsA and G proteins can be carried out to identify the targets and the outcomes of these studies may provide an expanded model for understanding G protein signaling in other human pathogenic fungi.”

Reviewer 2 Report

Dear Authors,

The manuscript ID: ijms-635876-v1 entitled “RgsA Attenuates PKA Signaling, Stress Response, and Virulence in the Human Opportunistic Pathogen Aspergillus fumigatus” written by Hnin Phyu Lwin, Yong-Ho Choi, Min-Woo Lee, Jae-Hyuk Yu and Kwang-Soo Shin contains a lot of interesting and important data about the rgsA gene predicted to encode a protein with an RGS domain alone.

Based on the very comprehensive research, the Authors showed that RgsA plays a multiple role in governing growth, development, stress response, virulence, and external polymer degradation. The article makes a some contribution to current knowledge.

The results are respectively presented in the form of figures such as photos, diagrams and schemes and properly interpreted. Manuscript is well written and organized.

I have no major comments on this paper, except some small suggestions, which are the following:

The section “Conclusions” is not mandatory, but can be added short separate summary to the manuscript.

Line 145: 2.5. Transcriptome Analysis: DEG Analysis and Functional Classification Please unify the font;

Line 332: × - X – Pleae unify the font in the whole text;

Line 325: Nucleic acid – Nucleic Acid

Best regards,

Author Response

Comments and Suggestions for Authors

Dear Authors,

The manuscript ID: ijms-635876-v1 entitled “RgsA Attenuates PKA Signaling, Stress Response, and Virulence in the Human Opportunistic Pathogen Aspergillus fumigatus” written by Hnin Phyu Lwin, Yong-Ho Choi, Min-Woo Lee, Jae-Hyuk Yu and Kwang-Soo Shin contains a lot of interesting and important data about the rgsA gene predicted to encode a protein with an RGS domain alone.

Based on the very comprehensive research, the Authors showed that RgsA plays a multiple role in governing growth, development, stress response, virulence, and external polymer degradation. The article makes a some contribution to current knowledge.

The results are respectively presented in the form of figures such as photos, diagrams and schemes and properly interpreted. Manuscript is well written and organized.

I have no major comments on this paper, except some small suggestions, which are the following:

The section “Conclusions” is not mandatory, but can be added short separate summary to the manuscript.

⇒ Thank you for your valuable comments! We added the section “Conclusions” (line 426-438).

Line 145: 2.5. Transcriptome Analysis: DEG Analysis and Functional Classification – Please unify the font;

⇒ Thank you for the comments! We unified the font (line 8, 106-107, 145, 416).

Line 332: × - X – Pleae unify the font in the whole text;

⇒ Thanks again! We unified the font in the whole text (line 80, 334).

Line 325: Nucleic acid – Nucleic Acid

⇒ Thanks again! We changed as your suggestion (line 327).